# RL Beats SFT while Mitigating Definition Bias in LLM-based Information Extraction

## Abstract

While large language models (LLMs) have been able to provide generally reasonable answers to complex information extraction (IE) tasks through prompt engineering and supervised fine-tuning (SFT), their performance and safety remain limited. We propose a novel *fuzzy matching* method to reveal that this is largely due to the *definition bias* between the model and the dataset. To mitigate this problem without human intervention, we use Reinforcement Learning with Verifiable Rewards (RLVR) to train the model, enabling it to independently learn the inherent definition of the task from the dataset. Specifically, we use Group Relative Policy Optimization (GRPO) to train LLMs of varying parameter sizes, rewarded with micro F1 scores, and achieve notably higher precision and recall than SFT across all models. We then apply fuzzy matching again to statistically demonstrate that this improvement is mainly primarily to the mitigation of the definition bias between the model and the dataset.

## 1 Introduction

In recent years, large language models (LLMs) have become a convenient solution for information extraction (IE) tasks (Xu et al., 2024). Due to their powerful generalization and instruction-following capabilities gained from their rich pre-training of general knowledge, current LLMs are already roughly capable of handling complex IE tasks. For example, consumer-level LLMs like GPT-4o OpenAI et al. (2024) can provide answers that human consider generally reasonable off-the-shelf. In addition, through prompt engineering and supervised fine-tuning (SFT), even much smaller LLMs, such as Qwen3-0.6B Yang et al. (2025), are able to generate generally reasonable responses.

However, while the model's answer may be correct in a general sense, it still falls short of the ground truth in specific scenarios. Even when the model recognizes the correct entity, it may under-extract or over-extract words around the entity, or classify the entity into a different category. For example, for text A in Table 1, the ground truth extracts "Apple" and classifies it as "organization", but the model may over-extract the "Inc." after it, or classify it into a different category like "location". Al-

A. Tim Cook is the CEO of Apple Inc.
  [PERSON]              [ORGANIZATION]

B. Marlowe Dynamics Inc., located at
   [ORGANIZATION]
The Virelli Tower, 30th floor, discloses the
              [LOCATION]
following information under this Agreement.

Table 1: Examples of texts with IE ground truths.

though the model's answer is more or less acceptable in general, it doesn't fully match the ground truth. This may cause serious consequences in some cases. For exmaple, when processing a confidential contract to extract and erase sensitive information in it, the model may under-extract or over-extract information, resulting in privacy leakage or unnecessary information loss. An example would be extracting organizations and locations from text B Table 1 for further erasure. Suppose we want to include floor numbers "30th floor" when extracting locations, but not "Inc." when extrating organizations. The problem is that no matter how good LLMs are at general language understanding, they may still fail to obey our rules, even after being trained on datasets carefully constructed according to our needs. As a result, it may not include "30th floor" but include "Inc.", which is the opposite of what we require, causing privacy leakage and unnecessary information loss respectively.

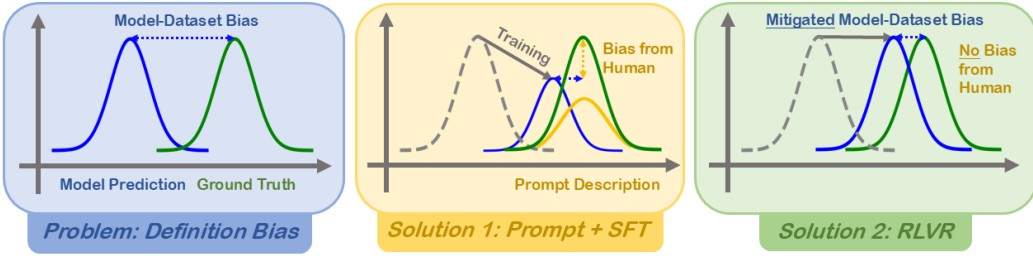

Figure 1: Diagrams of definition bias, the shortcoming of previous solutions and the advantage of RLVR. Green, blue and yellow curves represent the definition of the task implied by the dataset's ground truth, the model's prediction, and the human-designed prompt, respectively. Problem: the model and the dataset's definition differ, causing bias demonstrated as the distance between the blue and green curves. Solution 1: prompt designing with SFT mitigates definition bias (shorter horizontal distance), but also introduces extra bias from human (added vertical distance) after the model learns their definition. Solution 2: RLVR mitigates definition bias without introducing any new bias, since the model learns from the dataset by itself.

This is due to the *definition bias* between the model and the dataset we expect the model to predict (Huang et al., 2024), which we define as the gap between the model's understanding and the dataset's implied rules of the task. Since the model is pretrained and fine-tuned on general knowledge, it tends to solve tasks in a common way. However, due to industry norms or preferences, the dataset often specifies a task that differs from the most common scenario. This causes the model to generally understand the task but not strictly follow the dataset's rules. We further find that even if the model is fine-tuned on a training set with the same distribution, it still may not fully comprehend and conform to the dataset's definition of the task.

To alleviate this problem, a common practice is to write a clear and thorough system prompt for the model to reference, which may include an overall description of the task, definitions and restrictions for each category, extraction examples, etc., and use it in further supervised fine-tuning. Many relevant research have adopted this approach. Whether they design prompts one-off (Kwak et al., 2024; Neuberger et al., 2025) or refine them based on test results (Hein et al., 2025; Zhang et al., 2025), they share the same idea of manually designing sophisticated prompts for the model to follow.

While this approach is effective to some extent, it requires the system prompt to be designed by humans emperically. This introduces another bias between *humans* and the dataset, making it unable to completely solve the problem. In order to control the extra bias from humans, the system prompt needs to be precisely designed and constantly tested on every possible detail, which is time-consuming and laborious. Even so, since most datasets do not provide detailed rules for information extraction, it is still difficult to ensure the accuracy of the designed system prompt, thus preventing the human-introduced bias from being reliably mitigated.

Therefore, we require an approach that does not introduce extra bias from humans in the first place. In other words, we require the model to learn the inherent definition of the IE task from the dataset itself. Inspired by recent studies (Shao et al., 2024; DeepSeek-AI et al., 2025), we select Reinforcement Learning with Verifiable Rewards (RLVR) as our core approach. During reinforcement learning (RL), the model generates additional data to explore the dataset's implied rules, which are then scored by a rule-based reward function. By updating on self-generated positive and negative samples, the model learns the definition behind the dataset on its own. This avoids human-introduced bias from the start. It costs no manual system prompt design, and ensures that the model updates towards reducing definition bias. In Figure 1, we visually demonstrate definition bias, how previous methods introduce extra bias from humans, and how RL avoids human-introduced bias.

In this paper, we first discover how much impact definition bias has on model performance using a novel method, namely *fuzzy matching*. Then, we select Group Relative Policy Optimization (GRPO) (Shao et al., 2024) as our RL algorithm to train models of different parameter sizes on complex IE tasks. Afterwards, we compare the performances between the models trained with RL and SFT, and find that the former achieves better precision and recall under all parameter size settings. Finally, we apply fuzzy matching again to statistically show that such an performance gain is mainly due to

the mitigation of the definition bias between the model and the dataset, proving that RL effectively achieves our goal.

Our paper is organized as follows: In Section 2, we introduce *fuzzy matching* to evaluate the model's incorrect answers, and find that a large proportion of them results from definition bias, proving that definition bias seriously hinders model performance. In Section 3, we discuss the effectiveness of RL by designing a preliminary experiment to prove that RL enables the model to explore alternative solutions. In Section 4, we describe our training settings, including the datasets and training strategies. In Section 5, we conduct experiments to demonstrate that RL leads to better performance than SFT, and again use *fuzzy matching* to prove that the improvement mainly results from the mitigated definition bias between the model and the dataset.

## 2 SIGNIFICANCE OF DEFINITION BIAS

The examples in Table 1 have shown how the definition bias between the model and the dataset negatively impacts model performance. However, the extent of its impact remains to be estimated. We now explore the extent to which definition bias hinders model performance by measuring the improvement in model performance when definition bias is eliminated. If the improvement is large compared to the difference between perfect performance and the model's original performance, we conclude that definition bias is the primary factor contributing to the mediocre model performance.

Therefore, we design a *fuzzy matching* method to apply to the evaluation of the model's answers. For each extracted entity, we slightly relax the matching restrictions, and count answers that are "reasonable" but not exactly the same as the ground truth. If the results improve significantly, it indicates that definition bias is the primary factor hindering the model's performance. [1]

Specifically, we introduce two aspects in which fuzzy matching should be relaxed compared to *exact matching*. Firstly, when the model extracts the correct entity, it should be allowed to classify it into a category different from what the ground truth specifies. For example, "Harvard University" can be a "location" or an "organization" depending on one's view, so during fuzzy matching, the model is allowed to categorize the entity into either. Secondly, when the model extracts the correct core entity, it should be allowed to extract more or less words around the entity. For example, since extracting "Apple Inc." and "Apple" from the text "Tim Cook is the CEO of Apple Inc." are both generally acceptable, in this setting, both answers are considered correct. The number of mismatched words is defined as the *threshold*. See Figure 2 for more examples.

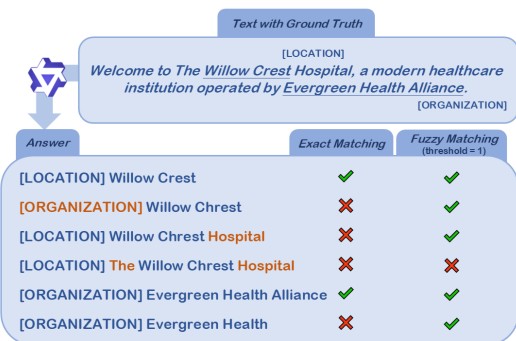

Figure 2: Differences between exact matching and fuzzy matching. Fuzzy matching allows the entity "Willow Chrest" to be classified into any category, including "location", "organization", etc. When the threshold is set to 1, fuzzy matching allows the LLM to over-extract or under-extract at most 1 word around the entity, such as "Willow Chrest Hospital" for "Willow Chrest" and "Evergreen Health" for "Evergreen Health Alliance", but not more than 1 word, such as "The Willow Chrest Hospital" for "Willow Chrest".

We select Qwen3-0.6B, Qwen3-1.7B, and Qwen3-8B (Yang et al., 2025) as our models, and perform SFT on them using the DWIE (Zaporojets et al., 2021) and DocRED (Yao et al., 2019) datasets. Then, we let the models generate answers to the questions in the test data. Afterwards, we apply exact matching and different degrees of fuzzy matching on them, calculate the micro F1 scores, and show them in Table 2. Finally, we calculate for all incorrectly extracted entities, what percentages of them can be fuzzy matched after each relaxation, and draw pie charts shown in Figure 3. From these

---

[1] Huang et al. (2024) have also introduced the concept of definition bias and two methods to measure it. However, these methods do not meet our requirements. See Appendix A for our detailed discussion.

statistics, we observe that with unlimited classification and a threshold of 2, models can improve 8.76%, 7.27% and 6.57% in preformance respectively, which are 43.37%, 49.59% and 51.45% of the distance to a 100% F1 score. This suggests that definition bias indeed exists, and is a large impediment to the model's performance.

| Matching Method | Qwen3-0.6B | Qwen3-1.7B | Qwen3-8B |
|---|---|---|---|
| Exact Matching | 79.80% | 85.34% | 87.23% |
| Unlimited Classification | 84.00% (+4.20%) | 88.48% (+3.14%) | 89.84% (+2.61%) |
| + Threshold = 1 | 87.45% (+7.65%) | 91.54% (+6.20%) | 92.84% (+5.61%) |
| + Threshold = 2 | 88.56% (+8.76%) | 92.61% (+7.27%) | 93.80% (+6.57%) |

Table 2: The average micro F1 score of models' answers on DWIE and DocRED when applying exact matching and different degrees of fuzzy matching. With unlimited classification and a threshold of 2, models can improve 6.57%-8.76% in the micro F1 score.

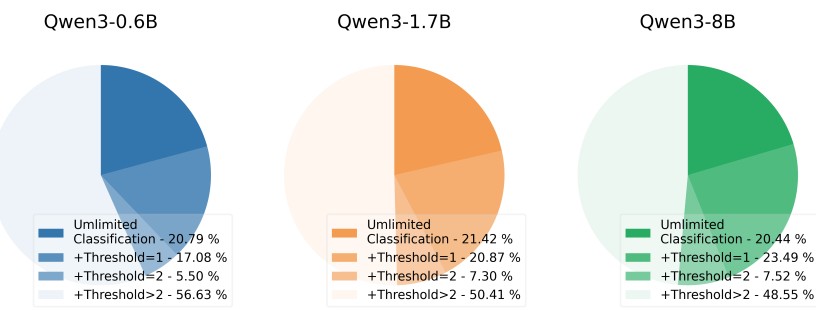

Figure 3: Pie charts showing the percentage of incorrectly extracted entities correctly that can be considered correct via fuzzy matching at each degree of relaxation. With unlimited classification and a threshold of 2, models improve by 6.57%-8.76% in the micro F1 score, which is 43.37%-51.45% from the original score to a 100% score.

## 3 EFFECTIVENESS OF REINFORCEMENT LEARNING

SFT aligns the model to output exactly what the dataset shows. Since it learns from a fixed number of samples, it fails to explore alternate interpretations that might better match the dataset's definition. This means the model's internal definition of "correct extractions" may remain misaligned, hindering the model's performance on the test set, even if token-level accuracy on the training set is high.

In contrast, RL frames extraction as an exploration–feedback process. The model first proposes an extraction under its current policy, and then updates the policy to maximize the expected reward. In this way, the model can learn from a wider range of samples generated by itself, and if the reward can reflect the degree to which the bias is mitigated, we expect the model to fit more accurately to the dataset's definition.

Previous studies (Shao et al., 2024; DeepSeek-AI et al., 2025) have proven the effectiveness of RL on general and mathematical tasks. To preliminarily investigate RL's ability to explore alternative solutions in our task setting, we perform RL on Qwen3-0.6B using the DpcRED dataset, and examine model generations that perfectly match the ground truth, but are textually different.

After RL, some cases are shown in Table 3. We see that although the model extracts the entities correctly, it may output them in a different order (in the first case), or output an entity multiple times in a category (in the second case), which is acceptable since they can be easily deduplicated afterwards. Therefore, the model has explored alternative solutions that are equally correct as the ground truth, and these solutions are also closer to the model's current output distribution, since they were generated by the model itself. Motivated by this, we now aim to conduct experiments further verify the effectiveness of RL on IE tasks.

| Model Answer | Ground Truth |
|---|---|
| [PERSON] Shakespeare; Anne; Terry; Jacques Rivette | [PERSON] Anne; Jacques Rivette; Shakespeare; Terry |
| [MISC] The Grim Adventures of Billy & Mandy; Evil Con Carne; Grim & Evil; The Grim Adventures of Billy & Mandy; "Cartoon Cartoons; Company Halt; Cartoon Cartoon | [MISC] Evil Con Carne; Cartoon Cartoon; The Grim Adventures of Billy & Mandy; Grim & Evil; Cartoon Cartoons; Company Halt |

Table 3: Some cases where the model generates a 100% correct answer that is textually different from the ground truth. In the first case, the model extracts all entites correctly, but in a different order. In the second case, the model extracts "The Grim Adventures of Billy & Mandy" once more than the ground truth, but should still be considered correct since the entities in a category can be easily deduplicated afterwards.

## 4 TRAINING SETTINGS

In this section, we select datasets and training strategies to train the model to compare its performance after SFT and RL on IE tasks.

### 4.1 DATASET SELECTION AND PROCESSING

For our main experiment, we select DWIE (Zaporojets et al., 2021) and DocRED (Yao et al., 2019) as the datasets, which consist of complex named entity recognition (NER) tasks. As shown in Appendix C, samples in these datasets consist of multiple sentences and a considerable number of words, thus requiring fairly powerful models to handle. In order to demonstrate the LLMs' ability to learn from different datasets simultaneously, we train the models on a mixture of these two datasets.

For each sample from the dataset, we add a system prompt before the text, which clarifies the source dataset, the categories along with their official descriptions copied from the original paper, and the output format, and then input it to the model.

In addition, we conduct supplementary experiments on relation extraction (RE) and entity extraction (EE) tasks. We use DWIE and DocRED for RE, and choose the DocEE (Tong et al., 2022) dataset for EE.

Statistics of the datasets, system prompts, and output format are shown in Appendix C.

### 4.2 TRAINING STRATEGIES

We train the same model with SFT and RL respectively to demonstrate that RL can lead to better performance of the model.

For RL, we choose Group Relative Policy Optimization (GRPO) (Shao et al., 2024) as our algorithm. During each step in GRPO, the model $\theta$ generates a batch of outputs $o_1, o_2, \ldots, o_G$ given the same input. Then, the reward function evaluates the responses and outputs their rewards $r_1, r_2, \ldots, r_G$. Their advantages are then calculated as the rewards normalized, and assigned to each token $t$, i.e.

$$\hat{A}_{i,t} = \frac{r_i - \text{mean}(\mathbf{r})}{\text{std}(\mathbf{r})} \quad (1)$$

Finally, the loss is calculated as follows:

$$\mathcal{L}_{\text{GRPO}}(\theta) = -\frac{1}{G} \sum_{i=1}^{G} \frac{1}{|o_i|} \sum_{t=1}^{|o_i|} l_{i,t} \quad (2)$$

where

$$l_{i,t} = \frac{\pi_\theta(o_{i,t}|q, o_{i,<t})}{[\pi_\theta(o_{i,t}|q, o_{i,<t})]_{\text{no grad}}} \hat{A}_{i,t} - \beta D_{\text{KL}}[\pi_\theta \| \pi_{\text{ref}}] \tag{3}$$

and used by the optimizer to update the model.

For the reward function, we design the following methods, and compare them in the preliminary experiment to find the optimal one:

1. Exact-match reward: use the micro F1 score with exact matching as the reward.

2. Span-aware reward: for each entity in the prediction ($Pred$), find the entity in the ground truth ($GT$) that has the most words overlapped with it. If the entity is classified into a different category in the ground truth, multiply the result by a factor $\alpha$. This process can be formulated as follows:

$$MaxOverlap(p) = \max_{g \in GT}(Overlap(p, g) \cdot (\alpha + (1 - \alpha)\mathbb{I}_{category})) \tag{4}$$

$\sum_{p \in Pred} MaxOverlap(p)$ is regarded as the number of true positives. The number of words in the prediction and ground truth are the number of positives and true samples, respectively. We use them to compute the F1 score as the reward.

3. Per-token reward: assign a reward for each token according to the following rules:

- if the answer contains redundant or incorrect entities, apply a -1 reward to the corresponding tokens;
- if the answer contains missing entities, apply a -1 reward to the token containing "]" at the end of the list;
- for other tokens, apply a +1 reward.

For example, with the ground truth being "locations: ['Paris', 'Berlin']" and the answer being "locations: ['London', 'Paris']", tokens "London" and "]" are given -1 rewards while the rest are given +1 rewards. Then, the rewards are then normalized to be advantages.

Additionally, through early experiments, we observe that directly applying GRPO to the model makes it difficult to converge to the required response format. To demonstrate this, we run an experiment directly applying GRPO to Qwen3-0.6B in Appendix B. Because of this, before GRPO, we slightly fine-tune the model using ground truths from the dataset, until it stably generates responses that follow the format.

We offer a comparison of training computation costs between SFT and RL in Appendix D.

## 5 EXPERIMENTS

### 5.1 EXPERIMENTAL SETUP

We compare RL (format learning + GRPO) with SFT (format learning + SFT) to demonstrate that the former produces greater performance gains. We select Qwen3-0.6B, Qwen3-1.7B and Qwen3-8B as our models, and run SFT and RL with the same total number of steps on our pre-processed dataset. During GRPO, the group size (i.e. the value of $G$ in Equation 2) is set to 8, and the length of responses are truncated to 512. After training, we use the precision, recall and micro F1 score to evaluate the performance of the models.

To find the optimal experiment settings, we conduct some preliminary experiments. First, to find the best-performing reward function, we apply each of the three reward functions designed on Qwen3-0.6B with DWIE and DocRED datasets, and evaluate the model's performance using the average precision, recall and micro F1. The results are shown in Table 4. We find that the exact-match reward function achieves the best F1 score, while the span-aware and per-token reward functions underperform in precision and recall, respectively. To investigate the reason, we observe the responses, and find that the span-aware reward function encourages the model to output an entity multiple times and in different categories, while the per-token reward function resulted in responses

exceeding the maximum length due to consecutively repeating the last entity or entities. Since the exact-match reward function performs the best, we use it in further experiments.

| Metric | Exact-Match | Span-Aware | Per-Token |
|---|---|---|---|
| Precision | **86.19%** | 53.01% | 85.50% |
| Recall | 83.78% | **86.45%** | 53.72% |
| F1 | **84.97%** | 65.52% | 65.21% |

Table 4: Performance of Qwen3-0.6B after RL with different reward functions measured by precision, recall and micro F1 on DWIE and DocRED. Exact-match reward achieves the best F1 score.

Next, to find a proper group size ($G$), we run GRPO on Qwen3-0.6B with $G = 4, 8, 16$ with DWIE and DocRED datasets, and evaluate the model's performance. The results are shown in Table 5. While the results of different settings of $G$ do not differ much, $G = 8$ achieves the best F1 score overall. Therefore, we set $G$ to 8 in subsequent experiments.

| Metric | DWIE | | | DocRED | | |
|---|---|---|---|---|---|---|
| | $G = 4$ | $G = 8$ | $G = 16$ | $G = 4$ | $G = 8$ | $G = 16$ |
| Precision | 88.19% | **88.73%** | 87.67% | 82.29% | **83.64%** | 83.40% |
| Recall | **86.70%** | 86.28% | 86.66% | 80.67% | 81.29% | **81.35%** |
| F1 | 87.44% | **87.49%** | 87.16% | 81.47% | **82.45%** | 82.36% |

Table 5: Performance of Qwen3-0.6B after RL with different number of generations per input ($G$) measured by precision, recall and micro F1 on DWIE and DocRED. While results of different $G$ settings are close, $G = 8$ achieves the best F1 score on both datasets.

## 5.2 BASIC RESULTS

The results of our main experiment on the DWIE and DocRED datasets are shown in Table 6. From the table, we observe that among all models, those after RL consistently perform notably better than those after SFT, with a micro F1 score increase of 2.38%-3.24% on DWIE and 1.46%-7.09% on DocRED. This indicates that using RL to train the model can lead to greater performance gains than SFT.

We also conduct an experiment on the full DWIE and DocRED splits. Due to computational limitations, we only run the experiment on Qwen3-0.6B. The results are shown in Appendix E.

For the additional experiments on RE and EE tasks, the results are shown in Appendix F.

| Metric | DWIE | | DocRED | | Average | |
|---|---|---|---|---|---|---|
| | SFT | RL | SFT | RL | SFT | RL |
| | | | Qwen3-0.6B | | | |
| Precision | 84.44% | **88.73%** (+4.29%) | 83.56% | **83.64%** (+0.08%) | 84.00% | **86.19%** (+2.19%) |
| Recall | 84.06% | **86.28%** (+2.22%) | 68.62% | **81.29%** (+12.67%) | 76.34% | **83.78%** (+7.44%) |
| F1 | 84.25% | **87.49%** (+3.24%) | 75.36% | **82.45%** (+7.09%) | 79.81% | **84.97%** (+5.16%) |
| | | | Qwen3-1.7B | | | |
| Precision | 86.05% | **91.25%** (+5.20%) | 85.63% | **86.44%** (+0.81%) | 85.84% | **88.84%** (+3.00%) |
| Recall | 87.47% | **88.77%** (+1.30%) | 82.30% | **84.36%** (+2.06%) | 84.88% | **86.56%** (+1.68%) |
| F1 | 86.75% | **89.99%** (+3.24%) | 83.93% | **85.39%** (+1.46%) | 85.34% | **87.69%** (+2.35%) |
| | | | Qwen3-8B | | | |
| Precision | 88.92% | **92.80%** (+3.88%) | 86.22% | **87.83%** (+1.61%) | 87.57% | **90.31%** (+2.75%) |
| Recall | 89.54% | **90.46%** (+0.92%) | 84.28% | **86.97%** (+2.69%) | 86.91% | **88.72%** (+1.81%) |
| F1 | 89.23% | **91.61%** (+2.38%) | 85.28% | **87.40%** (+2.12%) | 87.25% | **89.50%** (+2.25%) |

Table 6: Performance of SFT and RL measured by precision, recall and micro F1 on DWIE and DocRED. Models of different parameter sizes all achive better results after RL than after SFT.

## 5.3 CASE STUDY

To demonstrate the reason why RL performs better in the main experiment, we show some cases in the test set where the answer of the model after RL corrects the answer of the model after SFT in Table 7.

| Text with Ground Truth | Answer after SFT | Answer after RL |
|---|---|---|
| . . . Rhysently Granted won an open mic contest at the Southern Blues Bar . . . 
         [LOCATION] | [MISC] Southern Blues Bar | [LOCATION] Southern Blues Bar |
| . . . the lake that gave the municipality its name was drained in the early 20th century . . . 
           [TIME] | [TIME] the early 20th century | [TIME] 20th century |

Table 7: Some cases where RL outperforms SFT by mitigating definition bias. The first case shows that the model after RL corrctly classifies the entity "Southern Blues Bar" as "location", while the model afer SFT incorrectly classifies it as "misc". The second case shows that the model after RL correctly extracts "20th century", while the model afer SFT over-extracts "the early" before it.

**The model after RL classifies the extracted entity into the correct category.** In the first case, "Southern Blues Bar" is classified as "misc" (miscellaneous) by the model after SFT, and "location" by the model after RL. While these can both be considered correct depending on the scenario, the ground truths in the dataset always classify a bar as "location" instead of "misc", implying that the model after RL has a better understanding of the definitions implied by the dataset.

**The model after RL extracts the entity more accurately.** In the second case, when recognizing the century in the text, the model after SFT extracts "the early 20th century", while the model after RL extracts "20th century". Although both are reasonable answers, we scan through the DocRED dataset, and find that the ground truths never include "the early" before the century. Therefore, the answer of the mode after RL aligns better to the dataset's definition of the IE task.

## 5.4 EFFECTIVENESS OF RL IN MITIGATING DEFINITION BIAS

We now statistically prove that the improvement of each model after RL is mainly due to the reduced definition bias between the model and the dataset. To achieve this, we collect the entities that are correctly extracted by the model after RL but incorrectly extracted by the model after SFT, and count how many of them becomes correct due to reduced definition bias. Specifically, for each entity, we again apply different degrees of *fuzzy matching* to find out its counterpart in the answer given by the model after SFT. We first allow entities to be categorized into any category, and then gradually increase the number of mismatched words before and after the entity (i.e. the threshold), while counting the number of new entities that find their counterparts after each degree of relaxation. If most of the entities match a counterpart after slight relaxations, it indicates that the model after SFT is actually able to recognize most of these entities, but fails to extract them in the way the dataset does. Therefore, we can conclude that the difference in definition bias is the main contributor to the performance gap.

After counting the number of new matches after each degree of relaxation, we obtain a pie chart for each model shown in Figure 4. From the pie charts, we see that more than half (specifically, 51.04%-56.80%) of the entities in the RL model's answer after RL find their counterpart in the SFT model's answer after unlimited classification and no more than 2 mismatched words. This indicates that more than half of the performance improvement of RL is caused by the mitigation of the definition bias.

To demonstrate that fuzzy matching and splitting by a threshold of 2 is an effective attribution method, we randomly sample a subset of cases and perform human audit on them. See appendix G for details.

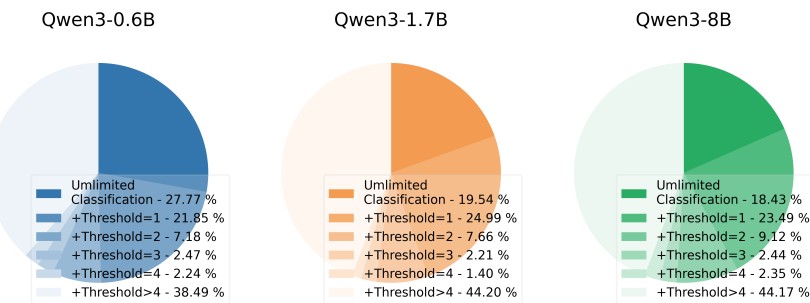

Figure 4: Pie charts showing the percentage of entities correctly extracted only by the model after RL that finds its corresponding entity in the answer of the model after SFT via fuzzy matching at each degree of relaxation. With unlimited classification and a threshold of 2, 51.04%-56.80% of the entities are corrected by the model after RL.

## 5.5 RESULTS ON SYNTHETIC DATASET

To further explore the effectiveness of RL in helping models learn implicit rules from datasets, we manually tweak DWIE and DocRED to synthesize a dataset that includes our own rules, and train Qwen3-0.6B with SFT and RL on it. Specifically, we add the following rules for each category:

- Location: enforce extractions of "in" before entities; disallow extractions of "on" or "at" before entities.
- Organization: enforce extractions of "Inc." after entities.
- Person: enforce extractions of "Mr.", "Mrs." and "Dr." before entities.
- Value: enforce extractions of "€" before entities; disallow extractions of "$" before entities.
- Misc: disallow extractions of "the" before entities.

After SFT and RL, we compute the recall of the entities related to each group of keywords.

The results are shown in Table 8. While for some keywords like "in" and "€", RL achieves the same recall or a slightly lower recall than SFT, for other keywords like "Inc.", "Mr. / Mrs. / Dr." and "the", RL achieves significantly higher recall scores, resulting in a higher score in average. This suggests that RL does help models learn implied rules from datasets.

| Category | Keywords | SFT | RL |
|---|---|---|---|
| Location | in | **86.86%** | 86.48% (-0.38%) |
| | on / at | 82.63% | **84.74%** (+2.11%) |
| Organization | Inc. | 63.17% | **83.91%** (+20.74%) |
| Person | Mr. / Mrs. / Dr. | 68.71% | **82.53%** (+13.82%) |
| Value | € | 100.00% | 100.00% (+0.00%) |
| | $ | 96.15% | 96.15% (+0.00%) |
| Misc | the | 61.68% | **72.55%** (+10.87%) |
| | | | |
| Average | | 79.89% | **87.67%** (+7.78%) |

Table 8: Recall of entities related to each keyword after SFT and RL on the synthetic dataset. RL achieves significantly better recall scores on adjusted entities in the "organization", "person" and "misc" categories, leading to a better average score compared to SFT.

## 6 RELATED WORK

**Definition bias between LLMs and datasets.** LLMs are already able to give generally resonable answers to IE tasks after SFT. However, there have been studies (Huang et al., 2024) that demon-

strate notable definition bias between LLMs and datasets regarding the IE task. While they showed that prompt engineering and SFT can mitigate the bias to a certain extent, they also stressed the complexity of creating comprehensive prompts to accurately describe the tasks. Proceeding from this, we systematically investigate the significance of definition bias, and show that by reinforcement learning, LLMs can comprehensively learn the dataset's definition of the task, and thus effectively mitigate the bias.

**Prompt engineering and SFT to mitigate definition bias.** Current studies often rely on prompt design to mitigate definition bias and improve the model's performance on IE tasks. Kwak et al. (2024) and Neuberger et al. (2025) manually design task descriptions, restrictions, extraction examples, etc. in one go, while the latter also adds detailed definitions of each category in the prompt. Hein et al. (2025) iteratively review the test results and manually refine the prompt to induce the desired behavior of the LLM. Zhang et al. (2025) start from human-designed prompts, and use LLMs to iteratively refine them based on test reseults. While these methods can mitigate the definition bias between the model and the dataset to some extent, they all require human intervention, which is laborious and introduces extra bias between humans and the dataset. In contrast, our method does not depend on the prompt. Instead, it lets the model learn the dataset's definition by itself, thus ensuring that no additional bias is introduced.

**Reinforcement learning for IE tasks.** Recent work has shown that reinforcement learning can substantially enhance LLMs' structured information extraction capabilities by providing verifiable, domain-aligned reward signals. Li et al. (2025) introduce MimicSFT and R²GRPO, combining template-guided supervision with relevance- and rule-based RL to improve scientific relation extraction and reduce reasoning errors. Similarly, Dai et al. (2025) leverage RLVR to train models to follow annotation-style reasoning procedures, yielding strong cross-domain robustness for relation extraction. While our methodology also includes applying RL on IE tasks, we focus more on diagnostics, exploring definition bias in IE tasks and how RL mitigates it.

## 7 CONCLUSION

Large language models (LLMs) are able to provide generally acceptable answers for information extraction tasks, but these answers may not follow the recognition logic implied by the dataset. In this paper, we use reinforcement learning (RL) with the micro F1 score as the reward to train LLMs to learn the implied definition behind the data on their own. Our experiments demonstrate that compared to supervised fine-tuning (SFT), RL achieves better results for all selected model sizes. By gradually loosening the restrictions when evaluating the RL model's answers, we statistically demonstrate that these performance gains are mainly due to the mitigation of the definition bias between the model's understanding and the dataset's inherent definition of the task.

ACKNOWLEDGMENTS

LLMs were used to polish writing, find proper datasets used in experiments (DWIE, DocRED and DocEE), and generate examples in Table 1 and Figure 2.

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

## A  INAPPLICABILITY OF THE MEASUREMENT METHODS FOR DEFINITION BIAS IN PREVIOUS WORK

While Huang et al. (2024) have also introduced the concept of definition bias with two methods to measure it, these methods do not meet our requirement. Their first method, *sentence similarty*, measures the bias between the pieces of text between datasets, rather than the bias between the dataset's ground truths and the model's answer. Their second method, *Fleiss' Kappa*, measures the difference between the answer and the ground truth based on exact matching, which can serve as a coarse-grained metric for the model's performance, but cannot distinguish the "close matches" which are caused by definition bias. For example, when the ground truth is "Apple Inc.", answers "Apple" and "Google" receive the same *Fleiss' Kappa* score, but the former reflects definition bias, while the latter is simply due to the model's poor performance.

## B  DEMONSTRATION OF PURE RL FAILING TO FOLLOW THE THE OUTPUT FORMAT

We directly apply GRPO to Qwen3-0.6B on DWIE and DocRED for 5 epochs, and show the average results in Table 9, compared to the method used in our main experiment (supervised format learning + GRPO).

| Metric | SFT | Format Learning + GRPO | Pure GRPO |
|---|---|---|---|
| Precision | 84.00% | 86.19% | 86.53% |
| Recall | 76.34% | 83.78% | 50.82% |
| F1 | 79.81% | 84.97% | 64.16% |

Table 9: Average performance of SFT, supervised format learning + GRPO and pure GRPO measured by precision, recall and micro F1 on DWIE and DocRED. GRPO without format learning performs poorly on recall.

We observe that the model outputs indented JSON objects instead of compact ones as the prompt asked, resulting in a waste of tokens. Worse still, it often (17.51% of the cases) outputs garbled text and/or consecutively repeat the last few tokens, leading to poor performance. For example:

```
{
    "location": ["India", "US", "Afghanistan", "Qatar", "Switzerland",
        "Mexico", "New Delhi", "Beijing", "South China Sea", "Japan", "
        Australia", "Vietnam", "India", "China", "Pakistan", "Isle
        Country Country Country Country Country Country Country Country
         Country Country Country Country Country Country Country
        Country Country Country Country Country Country Country Country
         Country Country Country Country Country Country Country
        Country Country Country Country Country Country Country Country
         Country Country Country Country Country Country Country
        Country Country Country Country Country Country Country Country
         Country Country Country Country Country Country",
    "Gvincial",
    "Gvincial",
    "Gvincial",
    Gvincial:
    Gvincial,
    Gvincial:
    .......
```

## C DETAILS IN DATASET SELECTION AND PROCESSING

Statistics of all datasets used in the experiments are shown in Table 10. For efficiency, instead of using the entire datasets, we randomly select samples from the original datasets to form the datasets for our experiments. Therefore, the sizes of some training and test sets in the table are the result of random selection, not their original sizes.

Specifically, for our main experiment using DWIE and DocRED, we randomly select 702 samples from the DocRED dataset to match the number of training samples in DWIE so that the model learns from them evenly. We use the full sets for the rest.

For the DocEE dataset, we only select samples whose events are related to "Famous Person", e.g. "Famous Person - Give a Speech", "Famous Person - Divorce", etc. The number of categories in DocEE includes the number of event types.

| Statistics | DWIE | DocRED | DocEE |
|---|---|---|---|
| Training set size | 702 | 702 | 1281 |
| Test set size | 100 | 1000 | 323 |
| Average number of sentences | 22.43 | 8.14 | 34.60 |
| Average number of words | 532.02 | 167.46 | 646.11 |
| Number of categories | 8 | 6 | 41 |

Table 10: Statistics of all datasets used.

The system prompt clarifies the following:

- The source dataset.

- The categories: location, organization etc., along with their descriptions copied from the original paper.

- The response format: a JSON object where each key is a category name and the corresponding value is a list of recognized entities.

For example, the system prompt for DWIE is as follows:

```
The user will provide you with a document from the DWIE dataset. From the
    document, extract all the entites of the following types:

location: entities referring to a particular geographical location.
organization: organizations such as companies, governmental organizations
    , etc.
person: entities referring to people in general such as politicians,
    artists, sport players, etc.
misc: miscellaneous entity types such as names of work of arts, treaties,
    product names, etc.
event: events such as sport competitions, summits, etc.
ethnicity: entity type used to identify different ethnic groups.
value: values in general such as time, money, etc.
other: includes the nominal variations of entity types (e.g., includes
    variations of country names such as ``German", which is a variation
    of ``Germany").

You should answer in the following JSON format: {"location": [...], "
    organization": [...], "person": [...], "misc": [...], "event": [...],
    "ethnicity": [...], "value": [...], "other": [...]}
```

Below is an example of a valid output:

```
{"location": ["White House", "United States", "Iraq", "Middle East", "
    Fallujah", "Washington, D.C"], "organization": ["Senate", "House of
    Representatives", "American Institute for Contemporary German Studies
    ", "Johns Hopkins University"], "person": ["George W. Bush", "Jackson
```

```
     Janes", "Nixon", "Reagan", "Clinton", "Saddam"], "misc": [], "event
    ": ["State of the Union", "Watergate", "Iran-Contra Affair", "World
    War II"], "ethnicity": [], "value": ["President", "Jan. 20", "
    Wednesday"], "other": ["Americans", "Iraqi", "American"]}
```

# D   COMPARISON OF TRAINING COMPUTATION COSTS BETWEEN SFT AND RL

We offer a computational cost comparison using standard analytic FLOP estimations widely used in LLM literature. Since FLOPs for Transformer blocks are deterministic, this method gives accurate relative compute without requiring runtime measurement. Under this formulation, SFT requires roughly $3F$ FLOPs per training sample (1 forward + 1 backward), whereas GRPO with group size $G$ requires $(G + 3)F$ FLOPs. For $G = 8$, RL therefore uses approximately 3.7x more compute per optimization step than SFT.

While RL appears to be more computationally intensive, we believe this is a reasonable cost. Although the raw F1 improvements appear modest (2–5%), given the task setting, their impact is significant due to reducing key errors such as missing a sensitive entity, over-extracting personally identifiable information, and misclassifying an entity that triggers downstream actions, especially given that current LLMs are already operating near a high baseline. Moreover, the additional cost of RL exists only during training. During inference, latency, memory footprint, and deployment cost remain identical to SFT. RL yields a one-time computational cost that produces a more robust, definition-aligned model without any inference-time penalty. Therefore, the performance gains are both practically meaningful and cost-effective in deployment settings.

# E   EXPREIMENTS ON FULL DWIE AND DocRED DATASETS

We conduct an additional experiment on Qwen3-0.6B using the full DWIE and DocRED training sets (702 and 3053 samples, respectively). The only difference between this experiment and our main experiment is the number of samples in the DocRED training set (3053 and 702, respectively).

The results are shown in Table 11. To our surprise, even with an uneven sample distribution favoring DocRED, DWIE's results are still higher than those in the original setting. We speculate that this is due to the similarity between the two datasets, making DWIE's low proportion of influence negligible.

| Metric | DWIE | | DocRED | | Average | |
|---|---|---|---|---|---|---|
| | SFT | RL | SFT | RL | SFT | RL |
| Precision | 85.05% | **89.11%** (+4.72%) | 83.29% | **87.09%** (+4.56%) | 84.17% | **88.10%** (+4.67%) |
| Recall | 83.75% | **87.16%** (+4.07%) | 76.23% | **86.82%** (+13.89%) | 79.99% | **86.99%** (+8.75%) |
| F1 | 84.40% | **88.12%** (+4.41%) | 79.61% | **86.96%** (+9.23%) | 82.00% | **87.54%** (+56.76%) |

Table 11: Performance of SFT and RL measured by precision, recall and micro F1 on the full DWIE and DocRED datasets. Models of different parameter sizes all achive better results after RL than after SFT.

# F   EXPREIMENTS ON OTHER TASKS

Table 12 shows the results on relation extraction (RE) and event extraction (EE). DWIE and DocRED datasets are used for RE, their results averaged, while DocEE is used for EE. The results demonstrate that RL still outperforms SFT on RE and EE.

# G   HUMAN AUDIT TO PROVE THE EFFECTIVENESS OF OUR ATTRIBUTION

We design a human audit to bound potential over-attribution. Specifically, we randomly sample 420 cases where the RL model gives the correct answer while the SFT model is judged incorrect under

| Model | Metric | DWIE+DocRED (RE) | | DocEE (EE) | |
|---|---|---|---|---|---|
| | | SFT | RL | SFT | RL |
| Qwen3-0.6B | Precision | 67.06% | **73.17%** | 48.42% | **50.70%** |
| | Recall | **56.33%** | 54.86% | 48.49% | **57.56%** |
| | F1 | 61.22% | **62.46%** | 48.46% | **53.91%** |
| Qwen3-1.7B | Precision | 71.36% | **77.05%** | 50.99% | **51.90%** |
| | Recall | 59.88% | **60.90%** | 50.49% | **59.47%** |
| | F1 | 64.99% | **67.90%** | 50.74% | **55.43%** |
| Qwen3-8B | Precision | 74.97% | **79.62%** | **54.79%** | 53.51% |
| | Recall | 63.73% | **64.98%** | 61.02% | **65.38%** |
| | F1 | 68.84% | **71.50%** | 57.74% | **58.85%** |

Table 12: Performance of SFT and RL measured by precision, recall and micro F1 on RE using DWIE+DocRED and on EE using DocEE. RL outperforms SFT on both tasks.

exact matching but receives credit under a fuzzy matching level (class-permissive only, ±1, 2, 3, 4 and >4-token span tolerance). Human annotators, blinded to the threshold level, assess whether each SFT extraction is (1) semantically correct and (2) a "reasonable" extraction of the entity. We then report the agreement rate between human judgments and fuzzy matching decisions for each threshold in Table 13

| Fuzzy Matching Level | Human-Validated Precision |
|---|---|
| Class-Permissiveness | 86.21% |
| Threshold=1 | 82.46% |
| Threshold=2 | 75.76% |
| Threshold=3 | 58.33% |
| Threshold=4 | 50.00% |
| Threshold>4 | 5.11% |

Table 13: Human-validated precision of fuzzy-matched SFT predictions at different relaxation thresholds. The results show high precision at class-permissive and ±1-token settings, with gradual degradation at ±2 tokens and substantial drops beyond that point, indicating where fuzzy matching begins failing in capturing "not precise but reasonable" extractions.

The results show high alignment at class-permissive and ±1-token settings, with gradual degradation at ±2 tokens and substantial divergence beyond that point. This confirms that fuzzy matching with our chosen thresholds do not substantially over-credit SFT outputs and provides an empirical bound on any residual over-attribution.

