# OpenReview forum: "RL Beats SFT while Mitigating Definition Bias in LLM-based Information Extraction"
_ICLR.cc/2026/Conference — Submitted to ICLR 2026_

### Official Review · Reviewer_W4eh · 2025-10-27

**Soundness:** 1
**Presentation:** 3
**Contribution:** 1
**Rating:** 2
**Confidence:** 4

**Summary:**

Information extraction used to be an important task for NLP. In this work, the authors claim that current LLM's IE solutions via prompt engineering / SFT is limited in terms of both performance and safety. They hence introduced a fuzzy match method to loose the constraints of exact match and demonstrated that using this metric, RLVR w/ GRPO outperforms SFT, with notable F1, precision & recall increase across all models.

While in the experiments, RLVR w/ GRPO does outperform SFT across different datasets and models, this conclusion is of no surprise. I think the authors' contribution: (1) propose a simple fuzzy matching method, i.e. allow category mismatch & allow 1-word-exception exact match and (2) RL > SFT is plain, normal and well-known, without sufficient technical ideas or insights.

**Strengths:**

1. The writing is good, easy to read.

**Weaknesses:**

1. While the authors claim that previous methods would cause privacy leakage or unnecessary information loss, their evidence is not convincing enough.
(1) privacy leakage: it should be avoided before the IE process,
(2) information loss: it can be reflected on the metrics, however, the authors do not include any baselines in their experiments. Further, the sota F1 reported in DWIE on NER with Joint + BERT is 87.7, higher compared to Qwen3-0.6B in this work (87.49). The lack of comparison with other methods would raise the concern about the effectiveness and contribution of the proposed method.
2. The contribution of this work can be summarized into two points: (1) propose a simple fuzzy matching method, i.e. allow category mismatch & allow 1-word-exception exact match, and (2) demonstrate RL > SFT, both of which, from my perspectives, are plain, normal and well-known, without sufficient technical ideas or insights.

**Questions:**

None

---

> ### Author Response · Authors · 2025-11-30
>
> Thank you for your thoughtful feedback. We will address each of your concerns with a point-by-point response.
>
>
>
> > Weakness 1.1: While the authors claim that previous methods would cause privacy leakage or unnecessary information loss, their evidence is not convincing enough. (1) privacy leakage: it should be avoided before the IE process, ...
>
> **Response:** We agree that upstream processes (e.g., data access control) are important for preventing privacy leakage. However, our statement refers specifically to privacy leakage caused *within* the IE step of redaction pipelines, not pre-processing failures.
>
> In many real-world applications, such as legal document sanitization, medical record redaction, and enterprise contract processing, the IE system *is* the mechanism responsible for identifying sensitive spans before redaction or transformation. In such pipelines, the raw document is available to the IE system by design, and it is the IE system's job to correctly identify sensitive spans, so that they can be removed or masked.
>
> Thus, leakage *during IE* is part of the standard threat model in downstream privacy-preserving workflows, not something that pre-processing can eliminate. Our examples illustrate this typical usage scenario: if the extraction model fails to match the dataset's definition (e.g., omitting "30th floor" or incorrectly including "Inc."), the redaction process will accordingly leak or remove unintended information.
>
> Therefore, our discussion is not about general privacy practices, but about the IE model's responsibility in ensuring correct redaction scope, which is precisely why definition bias matters in safety-critical IE applications.
>
>
>
> > Weakness 1.2: ... (2) information loss: it can be reflected on the metrics, however, the authors do not include any baselines in their experiments.
>
> **Response:** Our intention is not to argue that our method outperforms all possible baselines in minimizing information loss, but rather to show that a substantial portion of extraction errors exhibited by existing LLM-based IE systems arises specifically from *definition bias*, a type of error that conventional precision/recall/F1 metrics are unable to disentangle.
>
> Under exact matching, all mismatches (e.g., missing "30th floor" vs. predicting an entirely incorrect entity) are penalized equally, which obscures whether the model genuinely failed to understand the text or merely failed to adhere to the dataset's implicit labeling conventions. Our fuzzy-matching analysis is designed precisely to isolate this source of error and quantify how much of the observed “information loss” stems from misalignment with dataset definitions rather than from model incapability. The experiments comparing SFT and RL consistently show that RL reduces this specific component of error.
>
> Because our contribution is to identify and mitigate definition bias, rather than benchmark against diverse IE baselines, adding further baselines would not change the core finding: RL decreases definition-induced extraction inconsistencies that are not captured by standard metrics alone.
>
>
>
> > Weakness 1.3: Further, the sota F1 reported in DWIE on NER with Joint + BERT is 87.7, higher compared to Qwen3-0.6B in this work (87.49). The lack of comparison with other methods would raise the concern about the effectiveness and contribution of the proposed method.
>
> **Response:** Thank you for pointing out the SOTA F1 of Joint + BERT on DWIE. However, our work targets a different problem setting and therefore a different baseline is appropriate.
>
> Joint + BERT is a task-specific, token-classification architecture trained exclusively for DWIE, whereas our experiments use general-purpose LLMs that must handle long contexts, heterogeneous datasets, and generative extraction formats where definition bias naturally arises. Our contribution is not to surpass all prior IE models in absolute F1, but to diagnose and mitigate *definition bias*, a challenge unique to generative LLM-based extraction and not present in token-classification architectures.
>
> Thus, the relevant baseline is the same model trained with SFT rather than other IE systems. Across all model sizes, RL consistently outperforms SFT by a significant margin, and our fuzzy-matching analysis shows that over half of the improvement stems directly from reduced definition-bias errors. Therefore, the effectiveness and novelty of our method lie in mitigating definition bias rather than competing for overall SOTA F1 on DWIE.

---

> > ### Author Response · Authors · 2025-11-30
> >
> > > Weakness 2: The contribution of this work can be summarized into two points: (1) propose a simple fuzzy matching method, i.e. allow category mismatch & allow 1-word-exception exact match, and (2) demonstrate RL > SFT, both of which, from my perspectives, are plain, normal and well-known, without sufficient technical ideas or insights.
> >
> > **Response:** Thank you for raising this concern. We would like to argue that these two items are not “plain” when placed in the context of our research. Our paper is not simply "fuzzy matching" and "RL > SFT". Rather, it is a comprehensive exploration of *definition bias* and how to mitigate it, except that it treats fuzzy matching as a tool and RL as the method.
> >
> > Specifically, it involves:
> >
> > 1. Measuring definition bias: showing definition bias is a major failure mode via fuzzy matching.
> > 2. Attribution to definition bias: demonstrating that >50% of the improvement after RL is specifically due to correcting definitional mismatches.
> >
> > These findings provide actionable guidance: if practitioners want better model performance on IE than what SFT provides, they can (1) measure definition bias with our fuzzy matching method and (2) apply our RL recipe to reduce the bias. We believe this is a complete, practical contribution.
> >
> > In addition, we have added the following experiments and analyses to our paper to enhance novelty and broaden task scope:
> >
> > 1. We added an experiment using only RL, showing that it failed to learn the task and generate stable responses, and clarified the RL method we used in our experiment included a supervised format-learning process prior to RL.
> > 2. We conducted additional experiments using span-aware and per-token rewards, and compared their results with our initial method.
> > 3. We added experiments on the relation extraction task.
> > 4. We introduced a human audit for our fuzzy matching method to bound potential over-attribution.
> > 5. We added a comparison of training computation costs between SFT and RL.

---

### Official Review · Reviewer_N27d · 2025-10-28

**Soundness:** 3
**Presentation:** 3
**Contribution:** 2
**Rating:** 2
**Confidence:** 4

**Summary:**

This paper argues that SFT in LLM-based information extraction often inherits human-defined biases—rules and conventions embedded in prompts or annotations—causing models to miss the implicit definitions present in datasets. The authors term this gap definition bias and propose mitigating it through RLVR, using GRPO with micro-F1 rewards. A fuzzy-matching evaluation quantifies the bias and shows that RL consistently outperforms SFT across datasets, with most improvements stemming from reduced definition bias rather than memorization.

**Strengths:**

1. The paper has a clear problem definition and motivation. The concept of definition bias is intuitively introduced and visually explained (Figure 1). The paper articulates well why standard SFT cannot eliminate such bias because it introduces human-defined prompt bias rather than learning dataset-implied rules.
2. The experiments span multiple model sizes (0.6B–8B) and datasets (NER + EE).  The synthetic dataset experiment is particularly convincing. It clearly demonstrates that RL can learn implicit rules without human intervention.

**Weaknesses:**

1. The paper has Limited Novelty in Methodology.The paper primarily applies existing RLVR techniques (notably GRPO and micro-F1 rewards) to IE tasks, without introducing substantial algorithmic innovations or methodological insights. Similar RLVR IE frameworks have been explored in prior work [1][2], yet these are not sufficiently cited or contrasted.
2. Although RL slightly improves F1 scores, the overall gains (≈2–5%) may not justify the heavy computational cost of RL compared to SFT. Moreover, the approach is confined to entity and event extraction, leaving it unclear whether the method extends to more complex reasoning or relation extraction scenarios.

[1]Li, Ran, et al. "Beyond path selection: Better LLMs for Scientific Information Extraction with MimicSFT and Relevance and Rule-induced (R $^ 2$) GRPO."
[2]Dai, Runpeng, et al. "R1-re: Cross-domain relation extraction with rlvr."

**Questions:**

NA

---

> ### Author Response · Authors · 2025-11-30
>
> Thank you for your thoughtful feedback. We will address each of your concerns with a point-by-point response.
>
>
>
> > Weakness 1: The paper has Limited Novelty in Methodology. The paper primarily applies existing RLVR techniques (notably GRPO and micro-F1 rewards) to IE tasks, without introducing substantial algorithmic innovations or methodological insights. Similar RLVR IE frameworks have been explored in prior work, yet these are not sufficiently cited or contrasted.
>
> **Response:** Thank you for your valuable feedback. We have added these prior studies to **6 Related Work** in our revision. However, we do not intend to contrast our work with them in terms of algorithmic innovations or methodological insights. Instead, we believe our novelty lies in the exploration of *definition bias* in IE tasks and how RL mitigates it.
>
> Specifically, we first measure definition bias in **2 Significance of Definition Bias**, showing that definition bias is a major failure mode via fuzzy matching. Then, after we apply the RLVR method, we demonstrate in **6.4 Effectiveness of RL in Mitigating Definition Bias** that most of its performance gain results from mitigating definition bias, thus proving it to be a solid solution to our proposed objective, i.e. mitigating definition bias.
>
> We believe our findings are novel and useful in a empirical and diagnostic perspectives. For example, if practitioners find that SFT does not achieve their goal in model performance, they can (1) measure definition bias with our fuzzy matching and (2) apply our RL recipe to reduce the bias.
>
> In addition, we have added the following experiments and analyses to our paper to enhance novelty and broaden task scope:
>
> 1. We added an experiment using only RL, showing that it failed to learn the task and generate stable responses, and clarified the RL method we used in our experiment included a supervised format-learning process prior to RL.
> 2. We conducted additional experiments using span-aware and per-token rewards, and compared their results with our initial method.
> 3. We added experiments on the relation extraction task.
> 4. We introduced a human audit for our fuzzy matching method to bound potential over-attribution.
> 5. We added a comparison of training computation costs between SFT and RL.
>
>
>
> > Weakness 2.1: Although RL slightly improves F1 scores, the overall gains (≈2–5%) may not justify the heavy computational cost of RL compared to SFT.
>
> **Response:** Thank you for raising this concern. We respectfully argue that the gains observed are in fact substantial for IE, and that RL's cost is justified in this specific setting for the following reasons:
>
> 1.  In IE tasks, a 2–5% increase of micro F1 corresponds to a large reduction in critical errors. Unlike general NLP tasks, IE errors often correspond to missing a sensitive entity, over-extracting personally identifiable information, and misclassifying an entity that triggers downstream actions. Thus the observed increases represent meaningful, high-impact improvements, especially given that current LLMs are already operating near a high baseline (≈85–90%).
>
> 2.  The additional cost of RL exists only during training. During inference, latency, memory footprint, and deployment cost remain identical to SFT. RL yields a one-time computational cost that produces a more robust, definition-aligned model without any inference-time penalty. Therefore, the performance gains are both practically meaningful and cost-effective in deployment settings.
>
>
>
> > Weakness 2.2: Moreover, the approach is confined to entity and event extraction, leaving it unclear whether the method extends to more complex reasoning or relation extraction scenarios.
>
> **Response:** Thank you for raising this point. We have conducted additional experiments on relation extraction, and have added the results to **Appendix F**.

---

### Official Review · Reviewer_KoAr · 2025-10-31

**Soundness:** 2
**Presentation:** 3
**Contribution:** 2
**Rating:** 4
**Confidence:** 4

**Summary:**

The paper studies “definition bias” in IE—mismatch between a model’s implicit task understanding and a dataset’s labeling conventions—and proposes (i) a fuzzy-matching evaluation to estimate the share of errors attributable to definition bias and (ii) an RL with verifiable rewards (RLVR) training that directly optimizes micro-F1 via GRPO. On DWIE/DocRED (plus smaller WikiNEuRal/DocEE and a synthetic variant), RL consistently improves precision/recall/F1 over SFT for multiple Qwen3 sizes, and a post-hoc fuzzy analysis attributes a substantial portion of the gains to reduced definition bias. The training stack includes a brief “format learning” SFT warm-up, then GRPO with group size G=8, and evaluation with micro-F1. Experiments run on up to 4×A100.

**Strengths:**

1) Clear problem framing: “definition bias” is a practical failure mode for IE; the fuzzy-matching lens is a useful way to quantify its impact.

2) Simple, reproducible recipe: brief SFT for format control + GRPO with micro-F1 reward; the ablation on group size (G) is helpful.

3) Consistent gains: RL improves F1 across Qwen3-0.6B/1.7B/8B on DWIE/DocRED; case studies illustrate fixes in class choice and span length.

4) Attribution analysis: the second fuzzy-matching pass that explains where RL gains come from (relabeling/class-boundary/span tweaks) increases interpretability.

5) Synthetic-rule test: shows RL can internalize implicit labeling rules better than SFT on several keyworded constraints.

**Weaknesses:**

1) Task scope: main story is document-level NER/entity extraction; relation extraction and more complex IE pipelines are left for future work, so broader generality remains uncertain.

2) Reward design granularity: micro-F1 is assigned as a single scalar advantage per response; the paper notes token-specific credit assignment trials underperform, but deeper alternatives (span-level or structure-aware credit) are not explored.

3) Fuzzy matching validity: while practical, the relaxation choices (class-permissiveness and ±k token span tolerance) embed assumptions; it would help to test multiple fuzzy schemes (or human audit) to bound over-attribution to “bias mitigation”.

4) Limited methodological novelty: the core modeling advance is applying RLVR to IE. The fuzzy-matching component is valuable but primarily an evaluation/attribution tool rather than a modeling innovation.

**Questions:**

1) Can you report results on the canonical/full DWIE/DocRED splits (or provide calibration that trends hold when scaling sample sizes)?

2) Any evidence that bias mitigation carries over to relation extraction or event arguments within the same corpora?

3) For the fuzzy analysis, can you include a small human audit to bound over-attribution?

---

> ### Author Response · Authors · 2025-11-30
>
> Thank you for your thoughtful questions and constructive feedback. We will address each of your concerns with a point-by-point response.
>
> > Weakness 1: Task scope: main story is document-level NER/entity extraction; relation extraction and more complex IE pipelines are left for future work, so broader generality remains uncertain.
>
> **Response:** We appreciate your observation regarding the task scope. We have conducted additional experiments on relation extraction, and have added the results to **Appendix F**.
>
>
>
> > Weakness 2: Reward design granularity: micro-F1 is assigned as a single scalar advantage per response; the paper notes token-specific credit assignment trials underperform, but deeper alternatives (span-level or structure-aware credit) are not explored.
>
> **Response:** Thank you for highlighting the issue of reward granularity. In our revised manuscript, we have designed a span-aware reward function, and compared its results with per-response and per-token rewards. See **4.2 Training Strategies** and **5.1 Experimental Setup**.
>
>
>
> > Weakness 3: Fuzzy matching validity: while practical, the relaxation choices (class-permissiveness and ±k token span tolerance) embed assumptions; it would help to test multiple fuzzy schemes (or human audit) to bound over-attribution to “bias mitigation”.
> >
> > Question 3: For the fuzzy analysis, can you include a small human audit to bound over-attribution?
>
> **Response:** Thank you for your insightful suggestion. In the revised manuscript, we include a human audit designed to bound potential over-attribution. Specifically, we randomly sample 420 cases where the RL model gives the correct answer while the SFT model is judged incorrect under exact matching but receives credit under a fuzzy matching level (class-permissive only, ±1, 2, 3, 4 and >4-token span tolerance). Human annotators, blinded to the threshold level, assess whether each SFT extraction is (1) semantically correct and (2) a "reasonable" extraction of the entity. We then report the human-validated precision of the samples under each level as follows:
>
> | Fuzzy Matching Level | Human-Validated Precision |
> | -------------------- | ------------------------- |
> | Class-Permissiveness | 86.21%                    |
> | Threshold=1          | 82.46%                    |
> | Threshold=2          | 75.76%                    |
> | Threshold=3          | 58.33%                    |
> | Threshold=4          | 50.00%                    |
> | Threshold>4          | 5.11%                     |
>
> The results show high precision at class-permissive and ±1-token settings, with gradual degradation at ±2 tokens and substantial decrease beyond that point. This confirms that our chosen thresholds do not substantially over-credit SFT outputs and provides an empirical bound on any residual over-attribution. We have added these results to **Appendix G**.
>
>
>
> > Weakness 4: Limited methodological novelty: the core modeling advance is applying RLVR to IE. The fuzzy-matching component is valuable but primarily an evaluation/attribution tool rather than a modeling innovation.
>
> **Response:** Thank you for raising this concern. However, we would like to argue that our contribution is not merely a direct application of RLVR to IE. Rather, we believe our novelty lies in the exploration of *definition bias* and proof that RL mitigates it, which prior IE and RL research has not addressed.
>
> Specifically, we first measure definition bias in **2 Significance of Definition Bias**, showing that definition bias is a major failure mode via fuzzy matching. Then, after we apply the RLVR method, we demonstrate in **6.4 Effectiveness of RL in Mitigating Definition Bias** that most of its performance gain results from mitigating definition bias, thus proving it to be a solid solution to our proposed objective, i.e. mitigating definition bias.
>
> While fuzzy matching is indeed an evaluation/attribution tool, we argue that this does not mean that the work itself is not innovative. This tool is solidly used for measuring definition bias and demonstrating that RL mitigates it, contributing to a deeper exploration of definition bias in IE tasks, which we believe is novel and practical.
>
> In addition, we have added the following experiments and analyses to our paper to enhance novelty and broaden task scope:
>
> 1. We added an experiment using only RL, showing that it failed to learn the task and generate stable responses, and clarified the RL method we used in our experiment included a supervised format-learning process prior to RL.
> 2. We conducted additional experiments using span-aware and per-token rewards, and compared their results with our method using response-level advantage.
> 3. We added experiments on the relation extraction task.
> 4. We introduced a human audit for our fuzzy matching method to bound potential over-attribution.
> 5. We added a comparison of training computation costs between SFT and RL.

---

> ### Author Response · Authors · 2025-11-30
>
> > Question 1: Can you report results on the canonical/full DWIE/DocRED splits (or provide calibration that trends hold when scaling sample sizes)?
>
> **Response:** Thank you for raising this concern. We would like to clarify that in our original experiments, we have already been using the full training set of DWIE and the full test sets of DWIE and DocRED. The only difference lies in the DocRED train set, which we randomly sampled to match the number of training samples in DWIE so that the model learns from them evenly.
>
> We now conduct an additional experiment on Qwen3-0.6B using the full DWIE and DocRED training sets (702 and 3053 samples, respectively), and show the results below:
>
> |           | DWIE   |            | DocRED |            | Average |            |
> | --------- | ------ | ---------- | ------ | ---------- | ------- | ---------- |
> |           | SFT    | RL         | SFT    | RL         | SFT     | RL         |
> | Precision | 85.05% | **89.11%** | 83.29% | **87.09%** | 84.17%  | **88.10%** |
> | Recall    | 83.75% | **87.16%** | 76.23% | **86.82%** | 79.99%  | **86.99%** |
> | F1        | 84.40% | **88.12%** | 79.61% | **86.96%** | 82.00%  | **87.54%** |
>
> To our surprise, even with an uneven sample distribution favoring DocRED, DWIE's results are still higher than those in the original setting. We speculate that this is due to the similarity between the two datasets, making DWIE's low proportion of influence negligible.
>
> We have added this experiment in **Appendix E** in our revision. We have also added a statement in **Appendix C**, clarifying that for our main experiment, we only randomly select samples from the training set of DocRED, while using the full sets for the rest. We apologize for any confusion.
>
>
>
> > Question 2: Any evidence that bias mitigation carries over to relation extraction or event arguments within the same corpora?
>
> **Response:** Thank you for your thoughtful question. For relation extraction, we have conducted additional experiments on relation extraction within the same corpora, and have added the results to **Appendix F**. For event extraction, we note that DWIE and DocRED do not contain event-argument annotations, so this setting cannot be evaluated “within the same corpora.” However, we have conducted experiments on DocEE, a different dataset that does contain event-argument annotations, and have shown the results in **Appendix F**.
>
> > Question 3: For the fuzzy analysis, can you include a small human audit to bound over-attribution?
>
> **Response:** We have included a human audit to bound over-attribution, and have shown the results in **Weakness 3**. Please refer to that section.

---

### Official Review · Reviewer_brtf · 2025-11-01

**Soundness:** 2
**Presentation:** 2
**Contribution:** 2
**Rating:** 4
**Confidence:** 3

**Summary:**

This paper addresses "definition bias" in LLM-based information extraction. The authors propose using RL with Verifiable Rewards, specifically GRPO with micro F1 score as the reward, to enable models to learn extraction rules without human prompts. Authors introduce a novel "fuzzy matching" evaluation method to quantify definition bias by relaxing category and boundary constraints. Experiments on diverse Qwen3 models across DWIE and DocRED datasets show RL consistently outperforms SFT.

**Strengths:**

- The fuzzy matching approach is creative and provides an interpretable quantification of definition bias.
- RL improvements are consistent across all model sizes and datasets.
- The paper acknowledges negative results and discusses failure modes.

**Weaknesses:**

- The core contribution is applying existing RL techniques to IE tasks. While the application is novel, the technical contribution is incremental.
- The paper only compares RL vs. SFT. What about both SFT and RL?

**Questions:**

- A detailed training computational cost comparison should be provided.
- In Lines 480-481, the authors mention that "While we have also tried to assign token-specific advantages for finer granularity, the performance of the resulting model actually decreases". Could the authors provide more details on this part?

---

> ### Author Response · Authors · 2025-11-30
>
> Thank you for your valuable feedback. We will address each of your concerns with a point-by-point response.
>
> > Weakness 1: The core contribution is applying existing RL techniques to IE tasks. While the application is novel, the technical contribution is incremental.
>
> **Response:** Thank you for raising this concern. We agree that the "technical" contribution, i.e. applying RL to IE tasks, is incremental. However, rather than the application of RL, we believe the paper's novel contributions are empirical and diagnostic. Prior RL-for-IE work rarely, if ever (1) quantify how much of IE error stems from definition bias (we do, with fuzzy matching), and (2) explore the main reason for the improvement introduced by RL using multiple methods (case study, fuzzy matching and synthetic data), which consistently attribute the improvement to the mitigation of definition bias. Therefore, we believe our real novel contribution lies not in applying RL to IE tasks, but in proving that definition bias is the major obstacle to LLM-based IE, and that RL mitigates it significantly and robustly.
>
> In addition, we have added the following experiments and analyses to our paper to enhance novelty and broaden the task scope:
>
> 1. We added an experiment using only RL, showing that it failed to learn the task and generate stable responses, and clarified the RL method we used in our experiment included a supervised format-learning process prior to RL.
> 2. We conducted additional experiments using span-aware and per-token rewards, and compared their results with our initial method.
> 3. We added experiments on the relation extraction task.
> 4. We introduced a human audit for our fuzzy matching method to bound potential over-attribution.
> 5. We added a comparison of training computation costs between SFT and RL.
>
>
>
> > Weakness 2: The paper only compares RL vs. SFT. What about both SFT and RL?
>
> **Response:** We apologize for this confusion, and would like to clarify that the RL method we use in the experiments is actually already SFT + RL. At the end of **4.2 Training Strategies** in our initial manuscript, we have stated:
>
> > Additionally, through early experiments, we observe that directly applying GRPO to the model makes it difficult to converge to the required response format. Therefore, before GRPO, we slightly fine-tune the model using ground truths from the dataset, until it stably generates responses that follow the format.
>
> Now, to further investigate this, we directly apply GRPO to Qwen3-0.6B for 5 epochs, and show the average results of directly applying below:
>
> |           | SFT    | SFT * 2 + GRPO * 3 | GRPO * 5 |
> | --------- | ------ | ------------------ | -------- |
> | Precision | 84.00% | 86.19%             | 86.53%   |
> | Recall    | 76.34% | 83.78%             | 50.82%   |
> | F1        | 79.81% | 84.97%             | 64.16%   |
>
> We observe that the model outputs indented JSON objects instead of compact ones as the prompt asked, resulting in a waste of tokens. Worse still, it often (17.51% of the cases) outputs garbled text and/or consecutively repeat the last few tokens, leading to poor performance.
>
> In short, we observe that GRPO itself does not help the model learn the required response format quickly, nor does it stably generate responses. Therefore, we apply SFT for 2 epochs before GRPO to ensure the generated response conforms to the required format, thus guaranteeing the stability of the RL process. In our experiments, we control the total number of epochs in the SFT group (5) and the GRPO group (2 + 3) to be the same.
>
> We apologize, and have added our clarification with the above experiment results to **Appendix B** in our revised manuscript.

---

> > ### Author Response · Authors · 2025-11-30
> >
> > > Question 1: A detailed training computational cost comparison should be provided.
> >
> > **Response:** Thank you for raising this question. Following your suggestion, we add a computational cost comparison using standard analytic FLOP estimations widely used in LLM literature. Under this formulation, SFT requires roughly 3F FLOPs per training sample (1 forward + 1 backward), whereas GRPO with group size G requires (G + 3)F FLOPs. For G = 8, RL therefore uses approximately 3.7x more compute per optimization step than SFT.
> >
> > While RL appears to be more computationally intensive, we believe this is a reasonable cost. Although the raw F1 improvements appear modest (2–5%), given the task setting, their impact is significant due to reducing key errors such as missing a sensitive entity, over-extracting personally identifiable information, and misclassifying an entity that triggers downstream actions, especially given that current LLMs are already operating near a high baseline (≈85–90%). Moreover, the additional cost of RL exists only during training. During inference, latency, memory footprint, and deployment cost remain identical to SFT. RL yields a one-time computational cost that produces a more robust, definition-aligned model without any inference-time penalty. Therefore, the performance gains are both practically meaningful and cost-effective in deployment settings.
> >
> > > Question 2: In Lines 480-481, the authors mention that "While we have also tried to assign token-specific advantages for finer granularity, the performance of the resulting model actually decreases". Could the authors provide more details on this part?
> >
> > **Response:** Thank you for raising this question. In our attempt to assign different advantages to each token, we designed the following reward function for each token:
> >
> > * if the answer contains redundant or incorrect entities, apply a `-1` reward to the corresponding tokens;
> > * if the answer contains missing entities, apply a `-1` reward to the token containing `]` at the end of the list;
> > * for other tokens, apply a `+1` reward.
> >
> > For example, with the ground truth being `locations: ["Paris", "Berlin"]` and the answer being `locations: ["London", "Paris"]`, tokens `London` and `]` are given `-1` rewards while the rest are given `+1` rewards. Then, the rewards are then normalized to be advantages.
> >
> > We used this method on Qwen3-0.6B. The average results are shown below, compared with the RL method we adopted, i.e. one reward per sequence:
> >
> > |           | SFT    | GRPO with Per-Sequence Reward | GRPO with Per-Token Reward | GRPO with Per-Token Reward after Exclusion |
> > | --------- | ------ | ----------------------------- | -------------------------- | ------------------------------------------ |
> > | Precision | 84.00% | 86.19%                        | 85.50%                     | 85.50%                                     |
> > | Recall    | 76.34% | 83.78%                        | 53.72%                     | 83.45%                                     |
> > | F1        | 79.81% | 84.97%                        | 65.21%                     | 84.46%                                     |
> >
> > As shown in the table, assigning token-specific advantages results in significant performance drop compared to the RL method we adopted, which is mainly caused by the drop of the recall. After observing the generated answers, we find that many (specifically, 42.64%) of them exceeded the maximum length due to consecutively repeating the last entity/entities, for example:
> >
> > > ..., "Armenian", "Armenian", "Armenian", ...
> > >
> > > ..., "brevet brigadier general", "brevet colonel", "brevet brigadier general", "brevet colonel", ...
> >
> > We calculate the metrics again after excluding these cases, and show them in the last column of the table above. We see that they are now close to the results of GRPO with per-sequence reward. This indicates that this problem is the primary cause of the performance drop. We have added the details and results on this part to **4.2 Training Strategies** and **5.1 Experimental Setup**.

---

### Meta-Review · Area_Chair_kdRH · 2025-12-28

**Summary:**

In summary, all reviewers acknowledge that the paper addresses an important and underexplored issue, a.k.a“definition bias”, which defines the mismatch between a model’s implicit understanding of an information extraction task and dataset labeling conventions. The authors propose using reinforcement learning with verifiable rewards via GRPO and micro‑F1 rewards to reduce this bias. They also introduce a fuzzy matching evaluation to quantify and attribute errors to definition bias.

While reviewers find the problem well‑motivated and the experiments thorough, most feel the technical contribution is limited. Applying RLVR to IE with micro‑F1 rewards is considered a straightforward extension of existing techniques rather than a novel algorithmic advance. The fuzzy matching approach is an insightful diagnostic tool, but is viewed as an evaluation heuristic, not a modeling innovation.

Overall, reviewers agree the paper is clear, well‑written, and empirically careful, but methodologically incremental. Most rate it as marginally below the acceptance threshold. I appreciate the response from the authors, but I am recommending rejection, mostly because of its limited novelty.

**Reviewer Concerns:**

The major concern is with the novelty of the proposed methods. for example:
- “The core contribution is applying existing RL techniques… the technical contribution is incremental.” (Reviewer brtf)
- “Limited methodological novelty… fuzzy matching is an evaluation/attribution tool rather than a modeling innovation.” (Reviewer KoAr)
- “The paper primarily applies existing RLVR techniques… without introducing substantial algorithmic innovations or methodological insights.” (Reviewer N27d)
- “RL > SFT is plain, normal, and well-known, without sufficient technical ideas or insights.”(Reviewer W4eh)

However, this problem has not been properly addressed by the reviewers. For others:

Reviewer brtf commended the creative fuzzy‑matching evaluation and consistent RL improvements yet requested clarification on compute cost, training composition (SFT + RL), and token‑level reward performance. The authors’ rebuttal has addressed these points by adding a detailed FLOP‑based cost analysis, clarifying the SFT + GRPO training pipeline, and providing concrete per‑token reward results that explained the observed performance decline.

Reviewer KoAr raised concerns about limited task scope, fuzzy‑matching validity, and missing full‑dataset experiments. The rebuttal effectively handled most of these issues: additional experiments now cover relation extraction and complete DWIE/DocRED training sets, and a 420‑case human audit validated fuzzy‑matching reliability at small tolerance levels, but broader generalization to more complex information‑extraction pipelines and sensitivity analyses for multiple fuzzy schemes remain unaddressed.

Reviewer N27d similarly questioned the method’s limited algorithmic novelty and high training cost compared with modest F1 gains. The rebuttal justified the expense as acceptable for safety‑critical IE tasks and added relation‑extraction results, but novelty and cost–benefit concerns persist.

Reviewer W4eh criticized the lack of comparisons with stronger baselines such as Joint + BERT, the minimal technical innovation beyond “RL > SFT,” and weak improvement on privacy or information loss. The authors clarified that their focus is not to surpass SOTA results but to diagnose and reduce definition bias, which partly addressed the motivation issue but not the comparative or methodological gaps.

**Reviewer Scores:**

Given that the novelty is the major issue, I personally don't think the reviewers would have changed their scores, and in fact, none of them provided any comments on the update of the rating.

---

### Decision · Program_Chairs · 2026-01-26

Reject